# Mediating Effects of Diagnostic Route on the Comorbidity Gap in Survival of Patients with Diffuse Large B-Cell or Follicular Lymphoma in England

**DOI:** 10.3390/cancers14205082

**Published:** 2022-10-17

**Authors:** Matthew J. Smith, Bernard Rachet, Miguel Angel Luque-Fernandez

**Affiliations:** 1Inequalities in Cancer Outcomes Network, Department of Non-Communicable Disease Epidemiology, London School of Hygiene and Tropical Medicine, Keppel Street, London WC1E 7HT, UK; 2Department of Statistics and Operations Research, University of Granada, 18071 Granada, Spain

**Keywords:** diffuse large B-cell lymphoma, follicular lymphoma, mediation analysis, epidemiology, comorbidity, survival

## Abstract

**Simple Summary:**

There are inequalities in cancer survival between patients with or without comorbidities. The healthcare pathway (i.e., diagnostic route) of a patient is thought to explain some of these inequalities. We explore how much of the effect of comorbidity on survival of patients with diffuse large B-cell lymphoma (DLBCL) or follicular lymphoma (FL) is explained by the diagnostic route (i.e., emergency diagnosis). We used mediation analysis to separate the effect of comorbidity on survival from its effect through diagnostic route. We found that, for DLBCL and FL, emergency diagnosis accounted for 24% and 16% of the inequalities in survival between comorbidity groups within 12 months since cancer diagnosis. This proportion reduced over time and was small after 5 years of follow up. Comorbidities can complicate the diagnosis and management of patients with DLBCL or FL. Our results show that greater research is needed to ensure patients with comorbidities have a timely diagnosis and will help to reduce the inequalities in cancer survival.

**Abstract:**

**Background:** Socioeconomic inequalities in survival from non-Hodgkin lymphoma persist. Comorbidities are more prevalent amongst those in more deprived areas and are associated with diagnostic delay (emergency diagnostic route), which is also associated with poorer survival probability. We aimed to describe the effect of comorbidity on the probability of death mediated by diagnostic route (emergency vs. elective route) amongst patients with diffuse large B-cell (DLBCL) or follicular lymphoma (FL). **Methods:** We linked the English population-based cancer registry and hospital admission records (2005–2013) of patients aged 45–99 years. We decomposed the effect of comorbidity on survival into an indirect effect acting through diagnostic route and a direct effect not mediated by diagnostic route. Furthermore, we estimated the proportion of the comorbidity effect on survival mediated by diagnostic route. **Results:** For both DLBCL (*n* = 27,379) and FL (*n* = 14,043), those with any comorbidity, or living in more deprived areas, were more likely to experience diagnostic delay and poorer survival. The indirect effect of comorbidity on mortality through diagnostic route was highest at 12 months since diagnosis (DLBCL: Odds Ratio 1.10 [95% CI 1.07–1.13], FL: OR 1.09 [95% CI 1.04–1.14]). Within the first 12 months since diagnosis, emergency diagnostic route accounted for 24% (95% CI 17.5–29.5) and 16% (95% CI 6.0–25.6) of the comorbidity effect on mortality, for DLBCL and FL, respectively. **Conclusion:** Efforts to reduce diagnostic delay (emergency diagnosis) amongst patients with comorbidity would reduce inequalities in DLBCL and FL survival by 24% and 16%, respectively. Further public health programs and interventions are needed to reduce diagnostic delay amongst lymphoma patients with comorbidities.

## 1. Introduction

Non-Hodgkin lymphoma (NHL) is a heterogeneous group of malignancies, two of the most common types are diffuse large B-cell lymphoma (DLBCL) and follicular lymphoma (FL). For both DLBCL and FL, survival has steadily increased and, at 5 years since diagnosis, is substantially higher amongst those with FL compared to DLBCL [1,2,3]. However, there has been a differential increase in better health outcomes between patient characteristics, which has exacerbated deprivation and comorbidity inequalities in survival [1]. For other cancers, differences in comorbidity status partly explain the deprivation gap in survival yet inequalities remain [4]. Research has suggested that inequalities are partly due to the interaction between the patient, with certain characteristics, and the healthcare pathway (for example, accessing a GP appointment) [5].

The presence of a comorbidity impacts on a timely diagnosis, which then impacts on survival length [6]. For example, having a comorbidity that exacerbates the symptoms of lymphoma could hasten an appointment with a healthcare professional, potentially leading to an earlier diagnosis and a longer survival time. On the other hand, a comorbidity that obscures symptoms of lymphoma could delay an appointment with a healthcare professional, eventually leading to a shorter survival time. In other words, the effect of patient characteristics (e.g., comorbidity status) on survival is mediated by the access to the healthcare system (e.g., route to diagnosis).

Understanding the interaction between patients with certain characteristics and the healthcare pathway is crucial for enhancing public health policies. Quantifying the effect of comorbidity status on survival that is attributable to route to diagnosis is important for the healthcare system to investigate sources of inequity, contrast, and target routes to diagnosis, and allocate essential resources.

Applying conventional methodological approaches is limited to analyses that do not account for factors on the causal pathway. Mediation analysis has been developed to disentangle the effect of an exposure on an outcome that is mediated by another factor [7,8,9]. We aimed to mechanistically describe whether the impact of comorbidity on the probability of death is attributed to its direct effect on the chances of survival or rather its indirect effect that is mediated through the route of cancer diagnosis (i.e., emergency vs. elective).

## 2. Methods

### 2.1. Study Design, Participants, Data, and Setting

We used data from a retrospective population-based cohort study of patients diagnosed with DLBCL or FL between 1 January 2005 and 31 December 2013, followed up to 31 December 2015. DLBCL and FL diagnoses were made according to the International Classification of Diseases for Oncology (ICD-O), 3rd edition, based on codes C82.0–C85.9 (Appendix A) [10]. Patients entered the study on the date of their diagnosis and were followed up until death or administratively right censored at the 31st of December 2015 whichever occurred first.

Data was obtained from population-based cancer registries within the English National Cancer Registry and Analysis Service (CAS) [11] and linked to patient’s electronic health records from Hospital Episode Statistics (HES) [12]. CAS contains patient and tumour information (i.e., birth, diagnosis, and vital status dates) and sex, age at diagnosis, and ethnicity. HES data (for the period 2003 to 2015) was used for the assessment of comorbid conditions according to ICD codes (Appendix A) and contained clinical and administrative information. Using HES data, we assessed, retrospectively, the presence of any record of a comorbidity diagnosis for all patients with DLBCL and FL: certain comorbidities must be recorded even if they are not related to the reason for the hospital admission. In HES, the diagnostic fields are completed from admission and throughout the patient’s episode during secondary care. HES can include up to 20 different diagnostic codes within one episode: 1 main clinical code (indicating the reason for the admission), 19 secondary clinical codes, and up to 24 operation/procedural codes. Episodes are coded at admission and then each time a patient moves between different hospital units. We restricted the inclusion of comorbidity records to those diagnosed prior to 6 months, and up to 2 years, before the date of cancer diagnosis; this restriction aims to capture non-cancer related comorbidities and to minimise the introduction of selection bias [13].

### 2.2. Outcome, Exposure, and Other Variables

The outcome of this study was time since diagnosis up to death observed within (i) 1 year, (ii) 3 years given survival at 1 year, and (iii) 5 years given survival at 3 years. The main exposure was comorbidity status, and the mediator was route to diagnosis (i.e., emergency diagnostic route versus other). Based on data availability and clinical reasoning, we included age at diagnosis, sex, deprivation level and ethnicity as confounders.

*Comorbidity status* was classified according to the Royal College of Surgeons (RCS) Charlson score (Appendix A): an adapted score that reduces the number of relevant comorbidities (in comparison to the Charlson comorbidity score [14]) by removing a category (peptic ulcer disease) and groups diseases together (e.g., diabetes mellitus codes with or without complications are grouped into a single category). The score represents the count of comorbidities of a patient. Unlike the Charlson comorbidity score, the RCS Charlson score does not weigh the comorbidities: making the assumption that any comorbidity has the same impact on short-term mortality [15]. For the interest of the analytical approach, we dichotomised the score (no comorbidities vs. one or more comorbidities).

*Route to diagnosis* (NCRAS dataset), or *diagnostic route*, was originally recorded as one of eight routes to diagnosis [16]. Patients diagnosed on a ‘death certificate only’ were excluded to remove bias. There is no nationally recognised screening programme for NHL, thus no patients were diagnosed via a ‘screen-detected’ route. The remaining routes were dichotomised into a binary variable indicating whether the patient was diagnosed following an emergency or elective presentation: elective presentation consisted of patients diagnosed through two-week-wait, general practitioner referral, inpatient, or outpatient.

*Deprivation level* is based on the Lower Super Output Area [17] (*LSOA*) of residence of the patient at the date of cancer diagnosis. This is information is publicly available from the Office for National Statistics. An LSOA is a geographical location with a median of 1500 inhabitants. From the Index of Multiple Deprivation [18] (IMD), the income domain was classified into one of five quintiles based on the national distribution of ranked deprivation scores in the 32,844 LSOAs. Each patient was linked with one of the 209 Clinical Commissioning Groups (CCG) where their LSOA resides [19].

*Ethnicity* (HES dataset), due to data sparsity amongst ethnic minorities, was recorded as white or other.

### 2.3. Causal Diagram

The assumed causal relationships between the variables are shown in Figure 1. The main exposure, comorbidity status, causally influences the diagnostic route and death at a certain follow-up time. For simplicity in the graph only (i.e., not in the analysis), we group the baseline confounders (age at diagnosis, gender, ethnicity, and deprivation level) but note that they will not have the same level of effect on other variables, specifically diagnostic route, treatment, and mortality. The number of GP appointments represents the number of interactions between the patient and the primary care system. For other cancers, the number of GP appointments up to 4 months prior to diagnosis is associated with emergency presentation [20]. We structured the causal diagram from left to right in accordance with the assumed time frame in which these events are expected to occur. The omission of confounders that are unobserved, such as previous GP appointments, and unmeasured mediators, such as stage at diagnosis and treatment, represent our causal assumptions. For example, we assume that the number of GP appointments prior to diagnosis does not affect survival except through its effect on diagnostic route. For graphical illustration we include the unmeasured confounders of the mediator-outcome relationship, U. The omission of an arrow from comorbidity status to U represents our assumption that the effect of comorbidity status on survival acts solely direct, or indirect, through diagnostic route. We assume that there are no unmeasured confounders (U) for the (i) comorbidity-survival, (ii) route-survival, and (iii) comorbidity-route relationships. Additionally, we assume that (iv) the effect of comorbidity on survival is either direct or indirect through diagnostic route only. For example, this assumption states there is no unmeasured confounder for the route-survival relationship that is itself effected by comorbidity. Lastly, we assume that (v) there is consistency of a patient’s record of survival, such that survival is not altered if we set the comorbidity and route to the values they would naturally take.

To define the assumption of no unmeasured confounding, we define the potential outcome $Y_{i}\left(a\right)$ represent the value of Y if A were set to a for patient i=1, 2, …,n. Firstly, we assume no interference (i.e., the potential outcome for patient i does not depend on the comorbidity status, $A_{i}$, of patient i). Secondly, we assume consistency, such that for those who have comorbidity status A=a, their observed Y is the same as what it would have been had they had comorbidity status A=a via the hypothetical intervention. Furthermore, we assume conditional exchangeability, such that comorbidity status A is independent of each of the potential outcomes, conditional on the baseline confounders. Finally, since we used conditional survival time (i.e., survival at 5 years conditional on surviving the first 3 years after diagnosis), we assumed that censoring was non-informative during this time interval [21].

## 3. Statistical Analysis

### 3.1. Descriptive Statistics

We described the characteristics of DLBCL and FL patients, separately, using counts and proportions, and calculated the odds ratios of having at least one comorbidity along with Wald test *p*-values. The proportion of patients diagnosed with DLBCL, or FL, was graphed by diagnostic route, over comorbidity status, and stratified by deprivation level (i.e., least compared to most deprived). We then estimated 5-year net survival (for least and most deprived) DLBCL or FL patients for each comorbidity status and diagnostic route using a cohort approach (administratively censored at 31 December 2015) and the Pohar Perme estimator [22] in the Stata [23] package *stns* [24].

### 3.2. Natural Effect Estimates and Proportion Mediated

We examined what proportion of the comorbidity gap in survival was explained by diagnostic route amongst patients diagnosed with DLBCL or FL. As the outcome, exposure, and mediator are binary variables, we focused on the decomposition of the *total causal effect* (TCE) into the *natural direct*, and *indirect*, *effects* (i.e., NDE and NIE, respectively) [25]. The natural effects are calculated using the *gformula* Stata command [26]. To illustrate the decomposition, we first define the natural direct and indirect effects in terms of nested counterfactuals, $Y\left(a,M\left(a^{*}\right)\right)$, which indicates the outcome Y if A took the value of a and M took the value it would have taken if A took the value of $a^{*}$. Here, A relates to the presence of comorbidities (i.e., A = 1 for one or more comorbidities, A = 0 for no comorbidities) and M related to the diagnostic route (i.e., M = 1 for emergency vs. M = 0 for other diagnostic route). The direct effect is then the comparison of $Y\left(a,M\left(a^{*}\right)\right)$ to $Y\left(a^{*},M\left(a^{*}\right)\right)$, which measures the direct effect of changing the comorbidity status. The indirect effect is the comparison of $Y\left(a^{*},M\left(a\right)\right)$ to $Y\left(a^{*},M\left(a^{*}\right)\right)$, which measures the indirect effect of changing the diagnostic route. The total effect is the summation of the direct and indirect effects.

We first define the logistic regression model for the outcome Y with mediator covariables C
$$logit\left[E\left\{Y\left(a,M\left(a^{*}\right)\right)|C\right\}\right]=\beta_{0}+\beta_{1}a+\beta_{2}a^{*}+\beta_{3}C+\beta_{4}a\cdot a^{*}+\beta_{5}a\cdot C+\beta_{6}a^{*}\cdot C$$
this gives the NDE odds ratio
$$\frac{odds\left\{Y\left(a,M\left(a^{*}\right)\right)=1\;|\;C\right\}}{odds\left\{Y\left(a^{*},M\left(a^{*}\right)\right)=1\;|\;C\right\}}=exp\left\{(\beta_{1}+\beta_{4}a^{*}+\beta_{5}C)\left(a-a^{*}\right)\right\}$$
and the NIE odds ratio
$$\frac{odds\left\{Y\left(a,M\left(a\right)\right)=1\;|\;C\right\}}{odds\left\{Y\left(a,M\left(a^{*}\right)\right)=1\;|\;C\right\}}=exp\left\{(\beta_{2}+\beta_{4}a+\beta_{6}C)\left(a-a^{*}\right)\right\}.$$

Their product measures the total effect: $odds\left\{Y\left(a\right)=1\;|\;C\right\}/odds\left\{Y\left(a^{*}\right)=1\;|\;C\right\}.$

The proportion mediated (PM) captures what would happen to the effect of comorbidity status on mortality (i.e., by how much it would be reduced) if we were to disable the pathway between comorbidity status and diagnostic route (i.e., setting it to its natural value in the absence of comorbidity). The PM captures how much of the effect of comorbidity status on mortality is because of the effect of comorbidity on diagnostic route. On the risk difference scale, the PM is the ratio of the NIE to the TCE (i.e., $PM=\frac{NIE}{TCE}$). As the outcome is binary and the measure is the odds ratio, the ratio scale is used, but the PM is calculated using a transformation, such that
$$PM=\frac{OR_{NDE}\left(OR_{NIE}-1\right)}{\left(OR_{NDE}\times OR_{NIE}\right)-1}.$$

As the comorbidity gap in survival changes over time since diagnosis, the binary outcome (mortality) was stratified into death within (i) 12 months, (ii) 36 months given 12 months survival, and (iii) 60 months given 36 months survival. Analyses were performed on each of the three binary conditional survival outcomes. Since cause of death records are often unreliable or unavailable in population-based cancer registry data, the estimation of net survival in the relative survival setting reduces bias arising from background population mortality. However, the interpretation of net survival estimates within mediation analysis is not yet well understood; thus, we used a binary indicator of all-cause mortality within specific time periods. No patients were lost to follow up (i.e., there was no right censoring). The outcome (all-cause mortality) and the mediator (diagnostic route) were modelled using logistic regression. Missing records of diagnostic route (5.8%) were imputed using single stochastic imputation within the g-computation procedure, and all variables were included in the imputation model.

We used Stata v.17 (StataCorp, College Station, TX, USA) for statistical analysis. The code used for this analysis is provided for reproducibility at https://github.com/mattyjsmith/Proportion-mediated-NHL (accessed on 23 August 2022).

## 4. Results

### 4.1. Summary Statistics

Overall, 41,422 patients in England, aged from 45 to 99 years, were diagnosed with Diffuse Large B-cell lymphoma (*n* = 27,379) or Follicular lymphoma (*n* = 14,043), between 1 January 2005 and 31 December 2013 (Table 1). The prevalence of at least one comorbidity was 11.4% and 8.2% for DLBCL and FL, respectively. For both DLBCL and FL, those with comorbidities were those diagnosed at an older age and living in more deprived areas. For DLBCL, the probability of the presence of a comorbidity was lower amongst females. Emergency diagnostic route was more likely amongst those with comorbidity (both DLBCL and FL).

Amongst FL patients with any comorbidity, the more deprived the area, the more likely the patients were to have an emergency presentation (Figure 2), Amongst DLBCL patients with any comorbidity, there was no apparent trend in diagnostic route by deprivation level. For both DLBCL and FL patients without any comorbidity, the proportion of patients in each deprivation level were similar when comparing emergency and elective diagnostic routes.

Net survival differed between the least and most deprived for DLBCL and FL (Figure 3). Of those without comorbidity, the difference in survival at 1 year since diagnosis amongst the most deprived patients was 6.4% (71.3% vs. 64.9%) and 2.4% (94.1% vs. 91.7%) lower than least deprived for DLBCL and FL, respectively (Table 2). For both DLBCL and FL, the deprivation gap in survival was apparent from 1 year and remained similar through to 5 years since diagnosis, except for those with at least one comorbidity where there was no apparent deprivation gap through 5 years (Table 2 and Figure 3).

### 4.2. Natural Effect Estimates

**Total causal effect (TCE)**. The total effect of comorbidity on survival is the summation of the effects shown in Figure 1. Amongst those with comorbidity, for DLBCL, the odds of death within 12 months were 1.50 (95% CI: 1.39–1.61) times that of those without comorbidity; for FL, it was 1.71 times (95% CI: 1.45–2.02) (Table 3 and Figure 4). Over time, the comorbidity effect slightly increased for patients with DLBCL, however, for FL, the comorbidity effect was lowest at 3 years conditional on 1 year survival. At 5 years conditional on 3-year survival, the comorbidity effect remained strong at 1.57 (95% CI 1.39–1.79) and 1.62 (95% CI 1.38–1.90) for DLBCL and FL, respectively.

**Natural indirect effect (NIE)**. The indirect effect of comorbidity status through diagnostic route decreased as time since diagnosis increased for both DLBCL and FL (Table 3 and Figure 4). For both DLBCL and FL, the indirect effect was highest within 12 months since diagnosis (DLBCL: OR 1.10 [95% CI 1.07–1.13], FL: OR 1.09 [95% CI 1.04–1.14]) and gradually reduced through to 5 years since diagnosis (DLBCL: OR 1.01 [95% CI 1.00–1.02], FL: OR 1.00 [95% CI 0.99–1.01]).

**Proportion mediated (PM)**. For DLBCL and FL, the proportion mediated was highest within the first 12 months since diagnosis and decreased over time since diagnosis (Figure 5). Within the first 12 months since diagnosis, about a quarter (95% CI 17.5–29.5) and a sixth (95% CI 6.0–25.6) of the effect of comorbidity on survival was mediated by diagnostic route, for DLBCL and FL, respectively. With increasing time since diagnosis, the proportion of the comorbidity effect mediated by diagnostic route decreased to 1.3% (95% CI 0.0–2.8) and 0.3% (95% CI −1.0–1.5) after 5 years since diagnosis of DLBCL and FL, respectively.

## 5. Discussion

We aimed to estimate the proportion of the effect of comorbidity on survival that is mediated by diagnostic route amongst patients diagnosed with diffuse large B-cell (DLBCL) or follicular lymphoma (FL) in England.

Our results suggest that an elective (compared to an emergency) diagnostic route would reduce the odds of mortality within the first 12 months since diagnosis by 24% and 16%, for DLBCL and FL, respectively. This effect reduced over a longer follow-up time but was still apparent at 3 years since diagnosis, given these patients had survived 1 year. Patients with FL are often, at least initially, managed via watch-and-wait, whereas DLBCL is commonly treated with intensive immunochemotherapy, which may depend on the presence of comorbidities. The difference in the comorbidity effect between the two lymphoma subtypes (i.e., 24% vs. 16%) might be explained by the lack of information on treatment allocation.

The proportion mediated estimates how much of the total effect of comorbidity on survival operates through diagnostic route. If the proportion were large in our study, then the effect of comorbidity on survival would primarily be through the diagnostic route. However, in our study, since the proportion was small there may be other pathways through which comorbidity is acting on survival. Since diagnostic route is thought to be a process that identifies, and separates, comorbidity from cancer-related symptoms, a small value for the proportion mediated shows that comorbidity does not have a large effect on diagnostic route. Implying that, during the healthcare interaction (e.g., a general practitioner consultation), comorbidity symptoms are being identified as separate from cancer related symptoms for most patients. On the other hand, since there is still an ostensive value for the proportion mediated, this implies that comorbidity does influence diagnostic route for some patients, possibly for those with milder or obscure cancer symptoms.

Other pathways for the effect of comorbidity on survival could be through its effect on treatment allocation, quality and number of general practitioner examinations, or stage at cancer diagnosis. Firstly, information within the data was not available for the prevalence of treatment allocation (e.g., immunotherapies such as rituximab). Treatment allocation may explain little of the comorbidity gap in survival for patients with FL because these patients are managed via a watch and wait approach, unless they have a high-grade lymphoma. For DLBCL, the first line recommended treatment is immunochemotherapy (i.e., R-CHOP); however, a patient’s history of cardiac conditions might explain the comorbidity gap in survival because these patients are more likely to be allocated less intensive (i.e., less cardiotoxic) treatments (e.g., R-CVP). For example, patients at risk of cardiotoxicity (i.e., patients with underlying cardiac conditions) are likely to receive less intensive immunochemotherapies (e.g., combination chemotherapy without doxorubicin). Secondly, the quality of general practitioner examinations prior to cancer diagnosis is associated with the possibility of missed opportunities for early diagnoses, leading to a higher proportion of emergency diagnoses [27]. Information on the quality of the examination may explain the effect of comorbidity on diagnostic route thereby reducing the proportion mediated that was found in this study. Thirdly, it is possible that similar results would be obtained with stage at diagnosis as a mediator. Stage at diagnosis is closely associated with route to diagnosis; for example, patients diagnosed via emergency presentation are likely to have severe symptoms and an advanced cancer presenting with a late stage [28]. Lastly, completion of a treatment plan is crucial for optimal chances of survival. Performance status is known to be associated with the failure to complete the planned treatment of R-CHOP, and treatments are often made less intensive due to the toxicity [29]. In this study, information on performance status was not available; since this was not accounted for, performance status might explain the comorbidity gap in survival through treatment allocation and completion.

Our aim was to study the causal effect of comorbidity on survival whilst studying intermediate pathways, this made the application of causal mediation analysis a natural choice. Another strength of this study was the large population-based data incorporating all patients diagnosed with DLBCL or FL between 2005 and 2013. To our knowledge, this study is the first to disentangle the effects of comorbidity on survival of patients with NHL. The effect of diagnostic route on survival of patients with NHL is well known; emergency presentation is strongly associated with poorer health outcomes and worse survival probability [16,30]. Moreover, the effects of comorbidity on a patient’s diagnostic route are becoming clearer: the presence of comorbidities can either hasten the cancer diagnosis (due to the patient having more numerous interactions with the healthcare system) or delay the cancer diagnosis (due to comorbidities with similar symptoms) [31,32,33]. The multi-faceted interactions along a lymphoma patient’s pathway is not well known but this study highlights the need for further research into this. For other cancers, and in England, a study found that when using stage at diagnosis as a mediator for the deprivation gap in survival, stage explained little of the survival inequalities.^4^ This could suggest that including stage as a mediator in this study would not add further information beyond that of diagnostic route.

Comorbidities, though more prevalent in patients who are elderly or live in more deprived areas, have been shown to explain little in age- and deprivation-related inequalities in DLBCL or FL outcomes [34,35,36]. For policy purposes, further research could investigate how our findings from mediation analysis varies by deprivation (and age). More complex mediation analyses would however require more detailed information on individual clinical factors and system-level factors. Such factors may act as barriers to help-seeking behaviour, particularly among more deprived and elderly populations [33].

This study has its limitations. Firstly, we used an ecological measure of a patient’s deprivation level, which was geographically defined using the patient’s LSOA at the time of cancer diagnosis. Ecological bias could be present in this study because area-based income deprivation is possibly a poor predictor of individual income status [37]. However, since deprivation level was a confounder, misclassification is expected to have only a small influence on our conclusions in this study. Secondly, we did not include patients diagnosed through death certificates. It is possible that these patients could have had a short disease course and a more severe ill health, leading to an emergency presentation. For these patients, the length of time that they lived with the disease before death (i.e., their survival) is unknown, and including these patients would give an underestimate of the true survival probability. Moreover, there was a small proportion of patients diagnosed through DCO (DLBCL 0.8%, FL: 0.3%): including these patients would have a negligible effect on our results. 

We used hospital episode records (i.e., Hospital Episode Statistics data) to determine a patient’s comorbidity history. These records have universal coverage, allow for longitudinal linkage (ideal for cohort studies), and adheres to standardised coding practices [38]. These records are primarily collated for reimbursement purposes within the National Health Service rather than for research purposes and it is possible that some comorbidities are poorly recorded. However, administrative data (such as HES) has been suggested as the best available option, in comparison to clinical records, to ascertain comorbidity status for the Charlson comorbidity index [39,40].

Although diagnostic routes are well-defined, the process to identify the route is often complex [16]. A ‘route to diagnosis’ is a sequence of interactions between the patient and the healthcare system, but for analytical purposes are grouped into eight broad categories. This study focused on the comparison of emergency diagnostic route compared to other routes. There are close similarities in the qualitative definition between two-week-wait (TWW) referrals by a general practitioner and emergency route to diagnosis. For example, if HES records indicate an emergency route but a TWW record exists, then the TWW record takes priority if the emergency record date is more than 28 days prior to the decision to treat date. Given than comorbidity and deprivation can contribute to a delay in treatment, it is possible that some of these patients (who were recorded as TWW) should have been recorded as emergency diagnosis. Further studies could investigate this hypothesis by assuming different proportions of those patients diagnosed through TWW were emergency diagnosis, and measuring the proportion mediated accordingly.

In conclusion, our results show the effect of comorbidity status on survival of patients with lymphoma in England are partly explained by diagnostic route. Efforts to reduce diagnostic delay amongst patients with comorbidity would reduce DLBCL and FL survival inequalities by roughly 24% and 16% within the first 12 months since diagnosis, for DLBCL and FL, respectively. The proportion of the mediated effect reduces over time but is still apparent at 36 months since diagnosis. Public health programs could be redefined and implemented to reduce diagnostic delay amongst lymphoma patients with comorbidities. Further research should examine whether our findings differed in underserved areas.

## Figures and Tables

**Figure 1 cancers-14-05082-f001:**
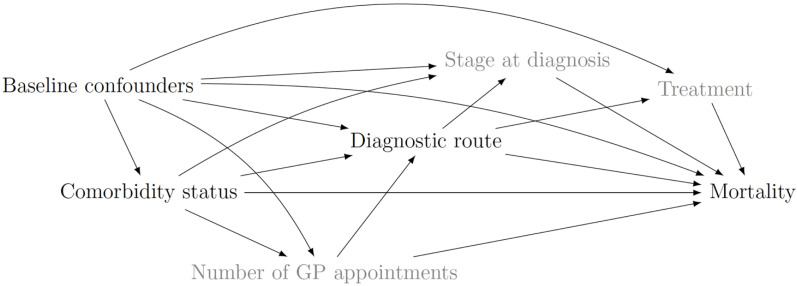
For patients diagnosed with DLBCL or FL in England between 2005 and 2013, this causal diagram represents the total effect of comorbidity on death mediated by diagnostic route, adjusted for baseline confounders (i.e., age at diagnosis, sex, deprivation, and ethnicity). Unmeasured confounders are stage at diagnosis, treatment, and number of general practitioner appointments.

**Figure 2 cancers-14-05082-f002:**
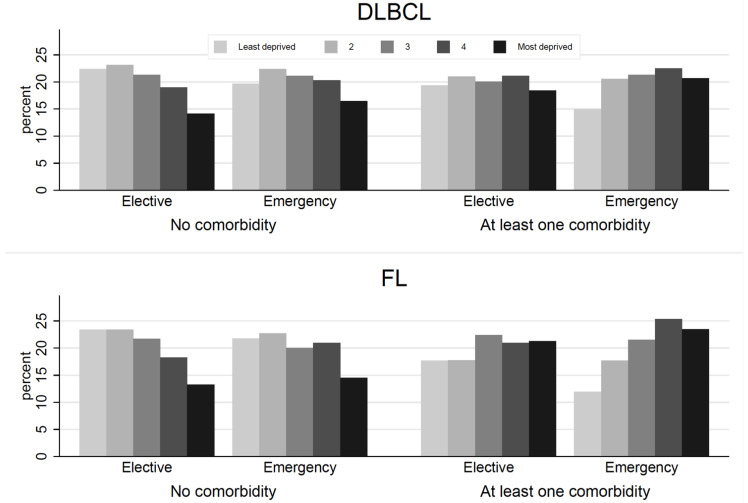
Distribution of deprivation levels by diagnostic route stratified over comorbidity status amongst patients diagnosed with DLBCL (*n* = 27,379) or FL (*n* = 14,043) in England between 2005 and 2013.

**Figure 3 cancers-14-05082-f003:**
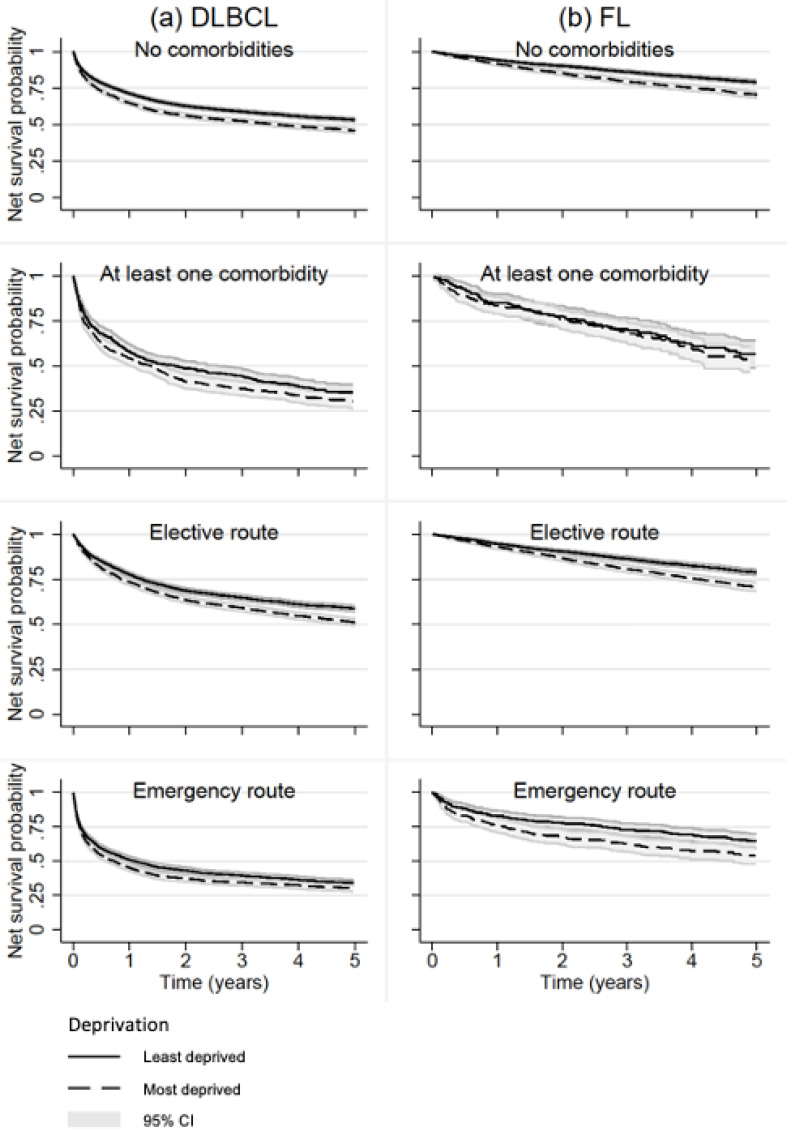
Net survival probabilities by comorbidity status and diagnostic route, stratified by deprivation level, amongst patients diagnosed with DLBCL (*n* = 27,379) or FL (*n* = 14,043) in England between 2005 and 2013.

**Figure 4 cancers-14-05082-f004:**
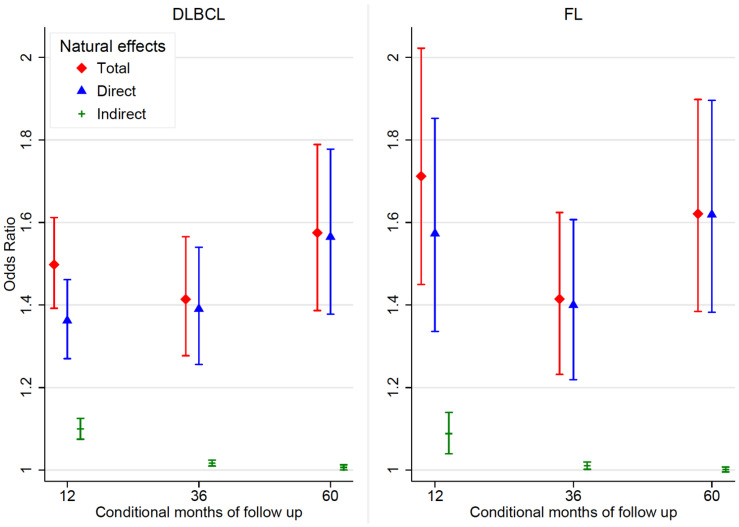
Effect of comorbidity status on odds of death at different conditional months of follow up since diagnosis amongst DLBCL (*n* = 27,379) or FL (*n* = 14,043) patients in England between 2005 and 2013.

**Figure 5 cancers-14-05082-f005:**
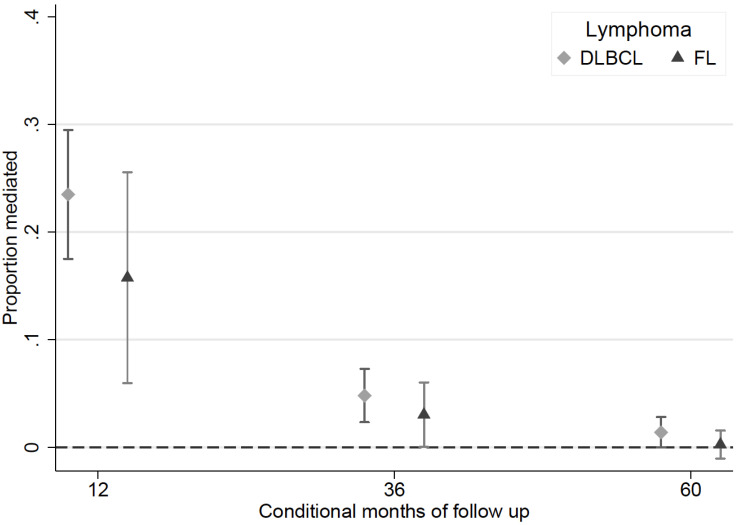
Proportion of the effect of comorbidity status on survival mediated by diagnostic route amongst patients diagnosed with DLBCL (*n* = 27,379) or FL (*n* = 14,043) in England between 2005 and 2013.

**Table 1 cancers-14-05082-t001:** Age at diagnosis, sex, deprivation level and ethnicity according to the comorbidity status amongst patients with Diffuse Large B-cell (*n* = 27,379) or Follicular lymphomas (*n* =14,043) in England between 2005 and 2013.

	No Comorbidity N (%)	Comorbidity N (%)	Total N (%)	OR ^†^ (95% CI)	*p*-Value *
**Diffuse large B-cell lymphoma**					
**Age *****					
Mean (SD)	70.3 (11.3)	74.1 (10.6)	70.7 (11.0)	1.35 (1.31–1.41) **	<0.001
**Sex**					
Male	12,904 (53.2)	1748 (56.2)	14,652 (53.5)	Ref	-
Female	11,365 (46.8)	1362 (43.8)	12,727 (46.5)	0.88 (0.82–0.95)	0.001
**Deprivation**					
Least deprived	5348 (22.0)	547 (17.6)	5895 (21.5)	Ref	-
2	5586 (23.0)	652 (21.0)	6238 (22.8)	1.14 (1.01–1.29)	0.031
3	5115 (21.1)	641 (20.6)	5756 (21.0)	1.23 (1.09–1.38)	0.001
4	4665 (19.2)	676 (21.7)	5341 (19.5)	1.42 (1.26–1.60)	<0.001
Most deprived	3555 (14.7)	594 (19.1)	4149 (15.2)	1.63 (1.44–1.85)	<0.001
**Route**					
Elective	15,495 (67.3)	1785 (58.8)	17,280 (66.3)	Ref	-
Emergency	7547 (32.8)	1252 (41.2)	8799 (33.7)	1.44 (1.33–1.56)	<0.001
Missing	1227 (5.1)	73 (2.4)	1300 (4.7)	-	-
**Follicular lymphoma**					
**Age *****					
Mean (SD)	66.2 (11.0)	72.0 (10.3)	66.7 (10.7)	1.62 (1.53–1.71) **	<0.001
**Sex**					
Male	5980 (46.4)	532 (46.5)	6512 (46.4)	Ref	-
Female	6918 (53.6)	613 (53.5)	7531 (53.6)	1.00 (0.88–1.12)	0.949
**Deprivation**					
Least deprived	3091 (24.0)	193 (16.9)	3284 (23.4)	Ref	-
2	3025 (23.5)	203 (17.7)	3228 (23.0)	1.07 (0.88–1.32)	0.487
3	2759 (21.4)	254 (22.2)	3013 (21.5)	1.47 (1.21–1.79)	<0.001
4	2356 (18.3)	253 (22.1)	2609 (18.6)	1.71 (1.42–2.09)	<0.001
Most deprived	1667 (12.9)	242 (21.1)	1909 (13.6)	2.32 (1.91–2.83)	<0.001
**Route**					
Elective	10,332 (87.2)	889 (81.0)	11,221 (86.7)	Ref	-
Emergency	1518 (12.8)	209 (19.0)	2407 (18.6)	1.60 (1.36–1.88)	<0.001
*Missing*	1058 (8.2)	47 (4.1)	1105 (7.9)	-	-

* Chi-squared test of association; ** Odds ratio for each 10-year increase in age; *** Range of 45 to 99 years; ^†^ Odds ratio of one or more comorbidities compared to none.

**Table 2 cancers-14-05082-t002:** Net survival estimates by comorbidity status and diagnostic route amongst patients diagnosed with DLBCL (*n* = 27,379) or FL (*n* = 14,043) in England between 2005 and 2013.

	1 Year NS (95% CI)		3 Years NS (95% CI)		5 Years NS (95% CI)	
	Least Deprived	Most Deprived	Least Deprived	Most Deprived	Least Deprived	Most Deprived
DLBCL						
Comorbidity						
None	71.3 (70.1–72.5)	64.9 (63.4–66.5)	58.9 (57.6–60.4)	52.5 (50.9–54.2)	53.3 (51.9–54.7)	45.7 (44.0–47.4)
At least one	58.0 (53.8–62.1)	54.2 (50.2–58.2)	44.1 (39.9–48.3)	37.4 (33.5–41.3)	35.4 (31.0–39.7)	30.4 (26.4–34.4)
Route						
Elective	77.8 (76.5–79.1)	73.6 (71.9–75.3)	64.8 (63.3–66.3)	59.2 (57.3–61.1)	58.8 (57.2–60.4)	51.1 (49.1–53.2)
Emergency	51.0 (48.6–53.4)	45.3 (42.8–47.8)	39.4 (37.1–41.8)	34.6 (32.2–37.0)	34.3 (32.0–36.7)	30.2 (27.8–32.6)
Follicular						
Comorbidity						
None	94.1 (93.3–95.0)	91.7 (90.3–93.0)	86.3 (85.0–87.5)	79.6 (77.7–81.6)	79.0 (77.5–80.5)	70.5 (68.2–72.9)
At least one	85.0 (80.0–90.0)	83.5 (78.8–88.2)	70.5 (64.0–77.0)	68.4 (62.5–74.3)	56.6 (48.9–64.3)	53.7 (46.8–60.7)
Route						
Elective	94.7 (93.9–95.6)	93.1 (91.9–94.4)	86.6 (85.3–88.0)	81.0 (79.0–82.9)	78.8 (77.1–80.5)	70.9 (68.4–73.3)
Emergency	82.8 (78.9–86.7)	75.9 (70.9–81.0)	72.9 (68.2–77.5)	62.7 (56.9–68.5)	64.2 (58.9–69.5)	53.5 (47.2–59.7)

DLBCL: diffuse large B-cell lymphoma; FL: follicular lymphoma; NS: net survival; 95% CI: confidence interval.

**Table 3 cancers-14-05082-t003:** Natural effect estimates for the odds ratio of conditional mortality since diagnosis, comparing comorbidity to no comorbidity mediated by diagnostic route, amongst patients diagnosed between 2005 to 2013 in England with DLBCL patients (*n* = 27,379) or FL patients (*n* = 14,043).

	1 Year OR (CI)	3 Years OR (CI)	5 Years OR (CI)
**DLBCL**			
TCE	1.50 (1.39–1.61)	1.41 (1.28–1.57)	1.57 (1.39–1.79)
NDE	1.36 (1.27–1.46)	1.39 (1.26–1.54)	1.56 (1.38–1.78)
NIE	1.10 (1.07–1.13)	1.02 (1.01–1.03)	1.01 (1.00–1.02)
PM	23.5% (17.5–29.5)	4.8% (2.3 –7.2)	1.3% (0.0–2.8)
**FL**			
TCE	1.71 (1.45–2.02)	1.41 (1.23–1.62)	1.62 (1.38–1.90)
NDE	1.57 (1.34–1.85)	1.40 (1.22–1.61)	1.62 (1.38–1.90)
NIE	1.09 (1.04–1.14)	1.01 (1.00–1.02)	1.00 (0.99–1.01)
PM	15.8% (6.0–25.6)	3.0% (0.0–6.0)	0.3% (−1.0–1.5)

NDE: natural direct effect; NIE: natural indirect effect; TCE: Total causal effect; PM: Proportion mediated; OR: odds ratio; CI: 95% confidence interval; DLBCL: diffuse large B-cell lymphoma; FL: follicular lymphoma.

## Data Availability

The data that support the findings of this study are available via application to the Public Health England Office for Data Release, but restrictions apply to the availability of these data.

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
