# Peer review of "Mediating Effects of Diagnostic Route on the Comorbidity Gap in Survival of Patients with Diffuse Large B-Cell or Follicular Lymphoma in England"

_cancers, 2022, doi:10.3390/cancers14205082_

Round 1
Reviewer 1 Report
This is an interesting and well-presented paper attempting to disentangle the role comorbidities play in diagnostic delay and the impact on outcome for two distinct types of lymphoma, one aggressive and potentially curable (DLBCL) and the other indolent and generally seen as a chronic disease (FL), but both of which can present with non-specific/vague symptoms leading to a delay in diagnosis.
The authors recognise the limitations of the study, the major one being not knowing whether the patient was treated with intensive chemotherapy, as this is obviously related to outcome. It is important to note that a large proportion of FL patients will be managed via watch and wait initially, despite some having extensive disease, this may explain why the co-morbidity effect was lower for FL. Despite the major limitation of lack of information on treatment, the methodology and information presented is novel and will potentially add a novel aspect to what is already known around diagnostic delay in lymphomas. However, if I have understood the analyses correctly a caveat should be added to the conclusion around diagnostic delay reducing survival by 24% and 16% respectively for DLBCL and FL amongst with this being the best-case scenario. If information had been available on treatment, these estimates would be different due to the strong relationship between whether a patient can be treated with intensive immunochemotherapy based on their comorbidities.
Minor comments:
Reference #3 needs defining properly in the reference list https://www.nature.com/articles/bjc201594#citeas
Page 2 methods, add "and FL" to the following sentence
"Using HES data, we assessed, retrospectively, the presence of comorbidities for all patients with DLBCL."
Figure 3 - if possible move legend to the bottom of the figure and include deprivation in the title
Author Response
Dear Reviewer,
Please kindly see our response to your comments in the attached file.
Kind regards,
Matthew J. Smith

Reviewer 2 Report
The paper examines the effect of the route to diagnosis on the association between comorbidity and survival from two of the common types of lymphoma- diffuse large B-cell lymphoma- an aggressive but curable form, and follicular lymphoma, which has a more indolent course. Lymphoma cases were identified from the English cancer registry (2005-2013), where the route to diagnosis was recorded as an emergency or not, and were linked to Hospital Episode Statistics to derive presence of any comorbidity 6 months before the cancer diagnosis. In first year of survival, the route to diagnosis accounts for reasonable proportion of the comorbidity effect on lymphoma survival, with little impact thereafter.
In the abstract, note that the electronic admission records were used rather than health records as the measure of comorbidity would be different depending on the source used to generate it. Give the period that cases where diagnosed (2005-2013). and the age range of the patients. The diagnostic route that accounts for the effect on comorbidity should be specified, that is an emergency compared to an elective route to diagnosis accounts for 24% and 16% of the comorbidity effect.
Comorbidity- please outline the diseases included in the RCS version of the Charlson Comorbidity Index and make clearer the differences from the original Charlson Comorbidity Index which is the score typically used in the study of DLBCL and FL outcome. Illnesses recorded in HES up to 6 months before diagnosis were included, but was there any early time limit e.g. 5 or 6 years pre-diagnosis - or 1 year for acute conditions- as diseases recorded closer to diagnosis would have more relevance than those that were more distant in time? On a similar note, were any recorded diagnosis included or were only those which were thought to be the reason for the hospital admission considered?
Deprivation was derived from the HES records- please clarify why was this necessary as the authors stated that deprivation was recorded on the cancer registry?
The causal diagram (Figure 1) implies that the baseline variables of age, sex, deprivation, and ethnicity have the same effect on other parts of the pathway. This may not be the case- for instance, age may directly influence whether a patient is treated or not, the other factors less so if at all. Others have suggested that these factors may not influence emergency presentation. Also, consider the terminology here as all other factors illustrated in the diagram- with the exception of treatment- would be considered confounders at baseline since they are determined at diagnosis. Lastly, the number of GP appointments are clearly important but only in the lead up to diagnosis- could some indication of a relevant timeframe be indicated in the figure?
In table 1, please add a total column so the characteristics of all patients with DLBCL and FL are described. Please state the proportion of patients with at least one comorbidity
Figure 2 is described as that among those with any comorbidity, the higher the deprivation category, the more likely that presentation as an emergency. This seems to be true for FL and less so for the more aggressive DLBCL; statistical tests here would aid interpretation for this and for comparisons amongst those with no comorbidity. For the comparison in the without comorbidity group, state what is being compared to be clear. In the text, it would be better to use the term “emergency presentation” rather than “diagnostic delay”.
Figure 3 does not show survival curves for the least and most deprived groups overall as the text suggests- please add. Also, it is stated that the deprivation gap in survival is present at 1y and widens at 3 and 5y for DLBCL and FL- however this pattern is not consistent across all graphs in Figure 3, nor in Table 2, as shown by the confidence intervals- please modify the text accordingly.
Please clarify the statements in the discussion regarding treatment allocation in the presence of comorbidity- the justification for treatment accounting for little of the comorbidity gap on the basis of universal healthcare is unclear and a comment on the efficacy of less intensive immunochemotherapies for those with cardiac conditions is warranted. Also, the association between stage and emergency presentation can be referenced as others have found this- other clinical markers eg ECOG, which is likely linked to comorbidity, have been linked to the diagnostic route and to survival but are not mentioned.
Limitations of the study could be extended. For instance, the authors could also comment on the impact of excluding DCOs on the findings- is it possible as these cases were not determined until death, could they have had a very short known disease course, are they more likely to have been patients that could have more severe ill health, and emergency presentation routes? The use of HES admission records for comorbidity also should be discussed- an inpatient stay is required and depends on what is recorded in terms of which conditions are noted in that stay, how many fields for conditions were used to construct the comorbidity measure; and also that HES records are primarily collated for reimbursement within the NHS rather than for research purposes.
Minor edits:
On page 5, please define the abbreviation PM if this is its first mention.
On page 5, it is stated that analyses were performed on 4 binary conditional survival outcomes but only three are listed in the previous sentence?
Figures 4 & 5 are repeats of the information in Table 3- consider whether the inclusion of the figures adds to the paper.
Author Response

(The authors gave the same response as above.)

Round 2
Reviewer 2 Report
Thank you for responding to my comments.